# How Central Carbon Metabolites of Mexican Mint (*Plectranthus amboinicus*) Plants Are Impacted under Different Watering Regimes

**DOI:** 10.3390/metabo13040539

**Published:** 2023-04-10

**Authors:** Lord Abbey, Raphael Ofoe, Zijing Wang, Sparsha Chada

**Affiliations:** Department of Plant, Food, and Environmental Sciences, Faculty of Agriculture, Dalhousie University, Truro, NS B2N 5E3, Canada

**Keywords:** edaphic stress, Jamaican thyme, Crassulacean acid metabolism, TCA cycle, Calvin cycle, nucleotide pathway, glycolysis, pentose phosphate pathway

## Abstract

Plants are sessile, and their ability to reprogram their metabolism to adapt to fluctuations in soil water level is crucial but not clearly understood. A study was performed to determine alterations in intermediate metabolites involved in central carbon metabolism (CCM) following exposure of Mexican mint (*Plectranthus amboinicus*) to varying watering regimes. The water treatments were regular watering (RW), drought (DR), flooding (FL), and resumption of regular watering after flooding (DHFL) or after drought (RH). Leaf cluster formation and leaf greening were swift following the resumption of regular watering. A total of 68 key metabolites from the CCM routes were found to be significantly (*p* < 0.01) impacted by water stress. Calvin cycle metabolites in FL plants, glycolytic metabolites in DR plants, total tricarboxylic acid (TCA) cycle metabolites in DR and DHFL plants, and nucleotide biosynthetic molecules in FL and RH plants were significantly (*p* < 0.05) increased. Pentose phosphate pathway (PPP) metabolites were equally high in all the plants except DR plants. Total Calvin cycle metabolites had a significantly (*p* < 0.001) strong positive association with TCA cycle (r = 0.81) and PPP (r = 0.75) metabolites. Total PPP metabolites had a moderately positive association with total TCA cycle metabolites (r = 0.68; *p* < 0.01) and a negative correlation with total glycolytic metabolites (r = −0.70; *p* < 0.005). In conclusion, the metabolic alterations of Mexican mint plants under different watering regimes were revealed. Future studies will use transcriptomic and proteomic approaches to identify genes and proteins that regulate the CCM route.

## 1. Introduction

A universal shift in weather patterns due to sustained global climate change has resulted in fluctuations in edaphic stress conditions, particularly drought, flooding, soil salinity, and soil impedance [1]. Plants are subject to such harsh and variable edaphic stress conditions because they are sessile. Typically, plant responses to the onset of edaphic stress include inhibition of plant root hydraulic conductance; reductions in stomatal conductance and photosynthesis; reprogramming of affected metabolic pathways; and, ultimately, cessation of plant growth and cell death [2,3,4,5,6]. The initial biochemical response to soil water stress is the production of free radicals and reactive oxygen species such as hydrogen peroxide and superoxide, which adversely affect plant morpho-physiological functions and interrupt metabolic processes to the detriment of the plant [7,8]. According to Stoychev et al. [9], such stress conditions lead to alterations in energy metabolism for survival and adaptation. 

The Calvin cycle occurs in the stroma of the chloroplast after photolysis (light reaction), and so it is referred to as the dark reaction phase of photosynthesis. From the light reaction, energy in the form of ATP and NADPH is generated and later used to produce glucose and downstream carbohydrate molecules in the Calvin cycle [10]. The carbohydrates from the Calvin cycle enter the central carbon metabolism (CCM) route via the glycolytic pathway [11,12]. The CCM is a complex series of enzymatic steps for the conversion of sugars into metabolic precursors in cells [7,8,13]. The CCM route, which consists of the glycolytic pathways, the pentose phosphate pathway (PPP) for sugar interconversion, and the tricarboxylic acid (TCA) cycle for final complete oxidation [7,14], provides 12 metabolites that are the basic carbon precursors of all biosynthetic pathways [15]. PPP is divided into an oxidative phase that produces ribose-5-phosphate for the synthesis of nucleic acids and a non-oxidative phase that produces NADPH [16]. The latter is an important reducing power for the synthesis of fatty acids, nucleotides, and non-essential amino acids for numerous metabolic pathways. So far, the impact of drought, flooding, or water stress recovery on CCM is understudied, but there are few reports on the individual pathways.

Cellular respiratory metabolism in plants via the TCA cycle occurs in the matrix of the mitochondrion, and it is essential for energy supply to different organelles for the maintenance of various physiological functions. A study conducted by Moradi et al. [6] revealed that drought disrupted the thyme (*Thymus* sp.) TCA cycle and reduced the synthesis of amino acids. In contrast, drought-induced metabolic reprogramming in Arabidopsis resulted in an increase in TCA cycle intermediate metabolites [17,18]. Kumar et al. [5] examined inbred maize (*Zea mays*) plants under water stress conditions and revealed that TCA cycle metabolites like succinate, α-ketoglutarate, and fumarate were reduced in drought-tolerant varieties while citrate and isocitrate were increased in drought-sensitive varieties. In addition, drought altered NADP+ concentration and electron receptor potential for the electron transport chain due to a reduction in NADPH+ and H+ availability for the TCA cycle. However, none of these studies reported on flooding or drought-flooding cycles. The ability of a plant to tolerate water stress is dependent on cellular concentrations of metabolic solutes like proline, betaine, fructose, and sucrose [19]. Research conducted using drought-tolerant wheat (*Triticum aestivum*) varieties showed that under severe water stress conditions, pyruvic acid, phenylpyruvate, fructose-6-phosphate, glucose, sucrose, and fructose were remarkably increased via the glycolytic pathway [3,5], suggesting an increase in sugar synthesis under water stress conditions. Nucleotide metabolism is the most critical cellular component for plant growth and affects many metabolic processes [20]. Purine and pyrimidine nucleotides are the building blocks for nucleic acid biosynthesis, which provides the required energy for the biosynthesis of carbohydrates, proteins, lipids, and secondary metabolites known to be central to all cellular metabolisms [21,22]. 

Water is crucial for plant growth and development, and globally, increasing numbers of farm operations are impacted by drought or flooding conditions because of changes in precipitation patterns caused by global climate change and competition with the rising global population and the manufacturing industries requirements for water [2,23]. Therefore, a fundamental understanding of plant response to fluctuations in water stress will be highly valuable for the development of stress-tolerant crops and the management of crop water requirements. Mexican mint (*Plectranthus amboinicus*) is an herbaceous perennial plant that belongs to the Lamiaceae family with a diverse array of ethnobotanical characteristics, culinary properties, and aroma-medicinal compounds [6,24]. Like most other plants, these properties can be affected by drought or flooding to varying degrees (Moradi et al. 2014 [6]). Although Mexican mint can survive drought conditions to some extent, its growth and chemical composition may be remarkably altered in prolonged stress conditions. 

Mexican mint is physiologically a Crassulacean acid metabolism (CAM) plant that fixes CO_2_ into C4 acids at night [25]. Because CAM plants absorb CO_2_ at night, their stomata are closed much of the day to conserve water and can tolerate limited drought conditions. However, the biochemical and physiological mechanisms underpinning the Mexican mint plant’s response to drought or flooding are not reported. Based on current knowledge, it was hypothesized that exposure of Mexican mint plants to prolonged drought or flooding will cause extensive disruption to CCM, but the plant will recover as soon as regular watering resumes. Therefore, the present study determined variations in metabolites of the different CCM routes following prolonged exposure of Mexican mint plants to drought or flooding and reversals to regular watering. This knowledge may be extended to other plants under the current dispensation of climate change and global warming.

## 2. Materials and Methods

### 2.1. Location

This research was conducted in the Plant Physiology and Stress Laboratory of the Department of Plant, Food, and Environmental Sciences, Faculty of Agriculture, Dalhousie University, and targeted metabolite quantitation was performed at the University of Victoria Genome BC (UVic GBC)—Proteomics Centre of The Metabolomics Innovation Centre, Canada, between December 2021 and April 2022. 

### 2.2. Preparation and Rooting of Cuttings 

Cuttings of soft tissue branches from the youngest second and third nodes on the main stem of healthy and well-watered Mexican mint mother plants were collected from the PFES greenhouse plant stock. Each cutting was trimmed to 5 cm in height with four pairs of corresponding leaves. Mexican mint is an easy rooting plant, so rooting hormone was not used. Four nodes on the stem part of the cuttings were embedded in moist perlite (Perlite Canada Inc., Montreal, QC, Canada) contained in a plastic flat tray of dimensions 50 cm length × 28 cm width × 6.5 cm depth. The planted trays were covered with a dome-shaped transparent plastic cover to maintain ≥95% relative humidity environment to induce rooting. The trays were placed on a planting shelf with 24 h fluorescence light at 22 °C. The cuttings were watered every other day, but the leaves were finely sprayed with water twice a day with no addition of fertilizer. The rooted cuttings were ready for transplanting after three weeks. 

### 2.3. Planting of Rooted Cuttings and Growing Condition 

Uniformly rooted cuttings were transplanted into 15 cm diameter plastic pots with saucers underneath, and each pot contained 200 g of Promix-BX mixed with 150 g of vermicast. The Promix-BX potting medium (Premier Horticulture Inc., Quakertown, PA, USA) contained 75–85% sphagnum peat moss, horticultural grade perlite, vermiculite, and dolomitic and calcitic limestone, and the vermicast was produced by Red Wiggler (*Eisenia fetida*) worms purchased from Growing Green Earthworm Castings (Lower Wedgeport, NS, Canada). The potted plants were arranged in a completely randomized design with four replications in a Biotronette Mark II Environmental Chamber (Lab-Line Instruments Inc., Melrose Park, IL, USA). The growth chamber was set at a 24°/20 °C day/night temperature cycle and a 12/12 h day/night light cycle. The plants were grown for 120 days, during which they were regularly watered to field capacity every three days prior to the imposition of water stress treatment. Pots were rearranged weekly to offset unpredictable occurrences due to variations in the environment.

### 2.4. Water Stress Treatment

To understand how the central carbon metabolism of plants recovering from water stress after reversal of drought (i.e., ≤10% field capacity) or flooding (i.e., oversaturation) is altered, the treatments used were regular watering (RW) at field capacity; continuous drought (DR); continuous flooding (FL); dehydration and resumption of regular watering after continuous flooding for 8 weeks (DHFL); and rehydration and resumption of regular watering after continuous drought for 8 weeks (RH). Regular watering was done every other day to maintain the field capacity of the growing medium. Flooding was simulated by complete submergence of the 15 cm diameter pot with the Mexican mint plant in water contained in a 20 cm diameter plastic pot. The experiment was arranged in a completely randomized design with four replications in the growth chamber.

### 2.5. Central Carbon Metabolites

#### 2.5.1. Sample Preparation

The green leaves of the potted Mexican mint plants were harvested after 8 weeks of water stress treatment and immediately dipped in liquid nitrogen (N), ground into a fine powder, and stored at −80 °C until analysis. The ground leaf tissue samples on ice were shipped by overnight courier to UVic GBC for targeted metabolite quantitation. Triplicate samples (50 mg) of ground leaf tissue per treatment were separately added to 500 L of 80% methanol and homogenized using a MM400 mill mixer (Retsch, Haan, Germany). The mixture was then sonicated for 5 min in an ice-water bath, followed by centrifugation at 21,000× *g* for 20 min at 5 °C. A quantity of 250 µL of the supernatants was removed and combined with 150 µL of dichloromethane and 150 µL of water. After centrifugal clarification of the mixture and vortex mixing for 30 s, triplicate samples of 80 µL aliquots of the supernatant were dried in a N gas flow. There were three biological replicates and two technical replicates. The obtained residues were used for the assays described below.

#### 2.5.2. Tricarboxylic Acid Cycle Assay

Standard stock solutions were prepared based on the method description provided by Han et al. (2013) [26]. The standard stock solutions of all targeted carboxylic acids were prepared in 80% methanol at concentrations of 200–1000 μM. A total of 50 μL of each standard solution or the supernatant of each sample was mixed and reacted with 50 μL of a 200 mM 3-nitrophenylhydrazines (NPH) solution at 30 °C. Following the reaction, 450 μL of water was added to each solution, and 10 μL of the resulting solution was injected into a C18 liquid chromatography (LC) column (2.1 × 100 mm, 1.8 μm) for quantification of the carboxylic acid by ultrahigh LC-multiple reaction monitoring/mass spectroscopy (UPLC-MRM/MS), and ion detection was performed on an Agilent 1290 UHPLC coupled to a Sciex 4000 QTRAP MS instrument (AB Sciex, Concord, ON, Canada) [26].

#### 2.5.3. Glucose and Selected Sugar Phosphates

An 80 L aliquot of the dried residue of each sample was mixed with 50 L of 50% methanol. After that, 50 mL of standard solutions of glucose, ribose, ribose-5-phosphate, glucose-6-phosphate, and mannose-6-phosphate were serially diluted before being combined and reacted with 100 mL of 25 mM 3-amino-9-ethylcarbazole (AEC) solution, 50 mL of 50 mM sodium cyanoborohydride (NaCBH_3_) solution, and 20 mL of LC/MS grade acetic acid for 70 min at 60 °C. A quantity of 300 μL of chloroform and 200 μL of water were added during the reaction period, after which each supernatant was centrifuged at 12,500× *g* for 5 min and vortexed for 15 s before adding 50 mL of water. Finally, 10 μL of the mixture was injected into a pentafluorophenylpropyl (PFP) LC column (2.1 × 150 mm, 1.7 μm) to run UPLC-MRM/MS on an Agilent 1290 UHPLC coupled to an Agilent 6495B QQQ instrument (Conquer Scientific, Poway, CA, USA) with positive-ion detection as previously described by [26].

#### 2.5.4. Other Metabolites

In 50% methanol, a solution known as an internal standard (IS) was created that contained 25 isotope-labelled metabolites including adenosine monophosphate (AMP), adenosine-5-triphosphate (ATP), uridine monophosphate (UMP), uridine triphosphate (UTP), uridine diphosphate (UDP)-glucose, fructose-6-phosphate (fructose-6P), fructose-bisphosphate, glycerol-3-phosphate, nicotinamide adenine dinucleotide (NAD), NADH, glucose-1-phosphate, ribose-5-phosphate, and others. Moreover, all the targeted metabolites were produced as serially diluted standard solutions in the IS solution at concentrations ranging from 0.00002 to 10 M. Eighty liter aliquots of dried residue were dissolved in 100 L of the IS solution. Ten microliters of each sample solution or each standard solution was injected onto a C18 LC column (2.1 × 100 mm, 1.9 μm) to run UPLC-MRM/MS and negative ion detection on a Waters Acquity UPLC system coupled to a Sciex QTRAP 6500 Plus MS instrument using tributylamine acetate buffer-acetonitrile/methanol (1:1, v/v) as the mobile phase with a gradient of 0.25 mL/min (from 10% to 50% B over 25 min) and 60 °C.

### 2.6. Calculation and Statistical Analysis

Concentrations of the detected analytes in the above assays were calculated with IS calibration by interpolating the constructed linear regression curves of individual compounds using the analyte-to-internal standard peak area ratios measured from injections of the sample solution. All data obtained were subjected to a one-way analysis of variance (ANOVA) using Minitab version 21 (Minitab, Inc., State College, PA, USA). Tukey’s honestly significant difference post-test was used to separate treatment means at *p* ≤ 0.05. A multivariate statistical analysis of grouped compounds, two-dimensional principal component analysis (PCA), and hierarchical clustering for differential metabolism per group were constructed with Euclidean distance using XLSTAT version 2022.3 (Addinsoft, New York, NY, USA).

## 3. Result and Discussion

### 3.1. Plant Growth 

The effects of prolonged drought and flooding stresses on plant growth and productivity are well documented [13,27], but not much has been reported on stress reversal. The results of the present study revealed that although morphological recuperation of Mexican mint through rehydration of drought-stressed (RH) plants and dehydration of flooded (DHFL) plants can be fast and occur within 4 h, the overall recovery of total plant size and leaf greening similar to that of regularly watered (RW) plants can be delayed (Figure 1A). This was proven by the resumption of leaf cluster formation on RH and DHFL plants compared to the stressed, i.e., drought (DR) and flooded (FL) plants, respectively (Figure 1A). 

### 3.2. Total CCM Metabolites 

Plants produce large quantities of specialized metabolites, which are end-products of cellular regulatory activities [28], but the compositions of these metabolites can be highly modulated by stress conditions [5,6,29], as demonstrated in the present study (Figure 1B,C). Anecdotally, Mexican mint as a CAM plant is known to tolerate some extent of water deficit stress [25], but its tolerance level and the mechanistic response to prolonged and continuous water stress conditions are unknown up to now. To understand this mechanism, changes in metabolites involved in CCM under different watering regimes were assessed. A two-dimensional PCA biplot presented in Figure 1B explained ca. 72% of the overall variations in total metabolites. The first (F1) and second (F2) factors represent ca. 41% and 31%, respectively. The PCA revealed three distinct clusters that clearly discriminated between the different water treatments with respect to the total metabolites of the CCM routes. Based on the cluster formation, it seemed the metabolism of RH and DHFL plants was adjusted after 8 weeks of stress recovery and resumed normal growth and metabolism that was similar to the RW plants. These results suggested that, irrespective of the prolonged drought and flooding, the Mexican mint plants underwent metabolic reprogramming when regular watering was resumed, as previously reported for intermediate metabolites in flood-stressed clover (*Trifolium* spp.) [9], drought-stressed Arabidopsis [17], and corn [5]. Drought or flooding causes remarkable degradation of chlorophyll and a significant impairment of photosynthesis, in addition to increased osmotic adjustment and MDA content [7,27]. These may explain the yellowing and stunting of the DR and FL plants compared to the RW plants. 

### 3.3. Metabolites Profile of the Different CCM Routes 

The results showed that the abundance of intermediate metabolites involved in the CCM routes was significantly (*p* < 0.05) altered due to water stress (Table 1). During water stress, plants go through rapid metabolic adjustment to maintain proper metabolism as an adaptation mechanism [12]. The process of metabolic adjustment requires several regulatory mechanisms to mediate signaling between multiple metabolic pathways and to initiate alterations in the composition of metabolites throughout the various CCM routes [7]. Metabolic analysis in the present study revealed the presence of 68 putative metabolites in the Mexican mint plants, which were categorized into five carbon-mediated metabolic groups for Calvin cycle, glycolysis, TCA cycle, PPP, and nucleotide biosynthesis [14]. 

Remarkable shifts in total concentrations of intermediate metabolites and energy generation were noticed with variations in watering regimes (Table 1). It is reported that plant response to water stress begins with rapid production of reactive oxygen species that act as an alarm signal that triggers acclimatory-defense responses by specific signal transduction pathways [7,8]. Consequently, these reactions result in a series of metabolic reprogramming in the plant. This can explain the significant (*p* < 0.05) alterations in total metabolites of the different CCM routes and energy generation in the Mexican mint plants following water stress imposition and water stress recovery, as shown in Table 1 and explained further below. 

#### 3.3.1. Calvin Cycle Intermediates

The production of downstream carbohydrate molecules, including glucose in the Calvin cycle, was significantly (*p* < 0.05) influenced by the different watering regimes (Table 1, Figure 2). Interestingly, the concentrations of total metabolites involved in the Calvin cycle were reduced non-significantly (*p* > 0.05) by ca. 15% in Mexican mint plants exposed to DR compared to those under RW (Table 1). On the contrary, the concentrations of total Calvin cycle metabolites were significantly (*p* < 0.001) increased by ca. 83% in FL plants compared to RW plants (Table 1). Plants that recovered from water stress, i.e., RH and DHFL plants, had their total Calvin cycle metabolites slightly increased by ca. 11% and 29%, respectively, although not significantly (*p* > 0.05) different from that of the RW plants (Table 1). This means that the water-stressed Mexican mint plants recovered their photosynthetic capabilities upon rehydration after a period of drought or upon dehydration after a period of flooding, and their photosynthetic capabilities approached those of the RW plants.

The ANOVA shows that the individual intermediate metabolites involved in the Calvin cycle varied significantly (*p* < 0.05) with variation in watering regime (Appendix A). Specifically, fructose-1,6-bisphosphate (F1,6BP) and ribulose-1,5-bisphosphate (RuBP) were significantly (*p* < 0.05) highest in DR plants. However, the overall reduction in total Calvin cycle metabolites in DR plants (Table 1) can be attributed to low concentrations of sedoheptulose-7-phosphate (Se7P), erythrose-4-phosphate (E4P), F1,6BP, dihydroxyacetone phosphate (DHAP), 3-phosphoglyceric acid (3PG), and ribulose-5-phosphate (Ru5P), as shown by the heatmap in Figure 2. The result suggested a possible reduction in the rate of photosynthesis in the DR plants, which was consistent with previous reports for other plants where drought reduced the abundance of numerous Calvin cycle proteins and other associated enzymes involved in photosynthesis [30,31,32]. Although enzymatic activities were not investigated, Wingler et al. [33] suggested that the reduction in Calvin cycle metabolites can be ascribed to impairment in photorespiratory enzyme activities, which can affect photosynthesis under drought stress. In soybean (*Glycine max*), Chen et al. [34] revealed that drought stress repressed the expression of genes involved in the Calvin cycle and resulted in a substantial reduction in photosynthetic abilities. 

On the other hand, the increase in Calvin cycle metabolites in FL plants can be ascribed to increased concentrations of Se7P, E4P, F6P, ribose-5-phosphate (R5P), and Ru5P (Figure 2). Previous studies demonstrated that the ability to maintain photosynthesis was strongly associated with flood tolerance in several plant species [35,36]. It was found that under flooding conditions, most flood-tolerant plants switch from aerobic to anaerobic respiration to increase photosynthetic carbon production. Such a flood adaptation mechanism could have been one of the reasons for increased Calvin cycle metabolites in FL plants, as reported for *Kandelia candel* by Pan et al. [37]. Furthermore, Se7P produced from sedoheptulose-1,7-bisphosphate (Se1,7BP) through the action of the enzyme sedoheptulose-bisphosphatase is critical for RuBP regeneration. Therefore, the increase in Se7P concentration in FL plants probably indicated an increase in photosynthetic rate and carbohydrate synthesis, as previously noticed by [38]. RH and DHFL plants had high concentrations of DHAP and Glyceraldehyde-3-phosphate (G3P), similar to concentrations found in RW plants (Figure 2). Collectively, G3P and DHAP are termed triosephosphates and can interconvert via the catalyzation of isomerase [39]. G3P and DHAP are important intermediates for photosynthesis and glycolysis. 

Studies have shown that during glycolysis, ca. 0.05–0.3% of glucose is non-enzymatically transformed into methylglyoxal, which can be increased following plant exposure to stressful conditions [40,41,42]. The present finding suggested that G3P, a key end-product of photosynthesis, was increased in plants under regular watering (i.e., RW) or plants that resumed regular watering after stress (i.e., RH and DHFL) compared to DR or FL plants. Sugar accumulation in response to drought and flooding has been reported by many authors [43,44,45]. Glucose is produced at the end of the Calvin cycle, and therefore, the increase in concentrations of glucose in both DR (high) and FL (from moderate to high) plants (Figure 3) can be considered a biochemical strategy to supply substrates for high-energy generation through glycolysis [11] as the stressed plants switched to a survival mode [9].

#### 3.3.2. Glycolytic Pathway Intermediates

The 3C sugar from the Calvin cycle enters the glycolytic pathway. In this study, we observed that the enzymatic breakdown of glucose via glycolysis was significantly (*p* < 0.001) higher in DR plants followed by FL plants when compared to RW plants (Appendix A). Total glycolytic metabolites were increased by ca. 1448% in DR plants and ca. 376% in FL plants compared to those of the RW plants (Table 1). Specifically, the increase in glycolytic metabolites in DR plants can be attributed to high concentrations of glucose, F1,6BP, phosphoenolpyruvic acid (PEP), pyruvic acid (PA), NADH, and adenosine triphosphate (ATP) (Figure 3; Appendix A). Additionally, the increase in glycolytic metabolites in FL plants can be attributed to high concentrations of F6P, PEP, and PA. These observations suggested that PEP and PA levels were increased in Mexican mint plants under both drought and flood conditions. Because total glycolytic pathway metabolites were not significantly (*p* > 0.05) altered in RH and DHFL plants, we suggested that glucose metabolism was altered by DR and FL conditions but was normalized after the plants were exposed to regular watering conditions, as portrayed by the RH and DHFL plants, respectively. 

Numerous studies have consistently reported that stress imposed by drought and flooding conditions accelerates glycolysis as a strategy for providing energy for stress defense activation and adaptation. For instance, Guo et al. [3] reported that glucose, PA, and PEP concentrations in wheat were considerably increased in drought-tolerant genotypes. Similarly, glucose and other glycolytic intermediate metabolites accumulated in *Pinus painaster* [46], *Thymus vulgaris*, *T. Kotschyanus* [23], and *Lotus japonicus* [47] in response to drought. Moreover, water stress instigated “energy crises” through the impairment of oxidative phosphorylation of the mitochondrion, resulting in a drastic reduction of ATP production [48,49]. Researchers have shown that sugar accumulation plays a crucial role in carbon resource allocation and plant growth [50]. To survive the “energy crises” under such stressful conditions, tolerant plants increase their glycolytic influx by accumulating more glucose to produce sufficient ATP via glycolysis to maintain basic cellular functions and regenerate NAD^+^ to maintain the glycolytic flux [48]. Furthermore, it has been reported that pyruvate metabolism plays a critical role in plant water stress tolerance [3,37,51]. A previous study by Pan et al. [37] also showed that *Kandelia candel* plants accumulated high concentrations of PA under flooding due to increased activities of 6-phosphofructokinase (PFK) and pyruvate kinase (PK). Additionally, PA production from glycolysis can be directed to a recycled pool of NAD+ through the fermentation pathway and/or serve as a hub for the biosynthesis of amino acids, fats, and sugars [52,53]. In poplar (*Populus* sp.) plants, PA accumulation under flood conditions was strongly linked to amino acid biosynthesis as a flood tolerance mechanism [54]. Although the activities of these enzymes were not examined in the present study, the increased PA concentrations in DR and FL plants (Figure 3) were consistent with previous findings. This suggested that the glycolytic pathway was crucial for modulating energy metabolism, which can lead to increased amino acid production in *Mexican mint* plants under water stress. Nevertheless, further studies are required to examine the detailed mechanisms involved.

#### 3.3.3. Pentose Phosphate Pathway Intermediates

Changes in the cellular abundance of PPP intermediate metabolites have been reported to strongly affect several other metabolic pathways [55]. In the present study, it was found that the total concentration of PPP metabolites was not significantly (*p* > 0.05) different amongst FL, DHFL, RH, and RW plants, and their average of 168.93 nmol/g was ca. 189% more than that of the DR plants (Table 1). Comparatively, the concentration of total PPP metabolites in DHFL plants was non-significantly (*p* > 0.05) the highest, followed by FL plants, and moderate in RW and RH plants. The low concentration of total PPP metabolite in DR plants can be associated with significant (*p* < 0.001) reductions in G6P and E4P, although the concentrations of R5P and xylulose-5-phosphate (Xu5P) were significantly (*p* < 0.001) enhanced compared to the RW plants (Figure 4; Appendix A). On the contrary, there was an increase in total PPP metabolite concentration in the FL plants (Table 1) that can be ascribed to a significant (*p* < 0.001) increase in the concentrations of R5P, Ru5P, Se7P, E4P, and F6P (Figure 4). 

Similarly, increased concentrations of G6P, E4P, G3P, and F6P accounted for the high concentrations of total PPP metabolites in DHFL plants (Figure 4; Appendix A). It was established that PPP is primarily responsible for the major supply of NADPH for several biosynthetic pathways in cells and contributes to antioxidant production [56]. The increased NADPH in DR plants possibly indicated an enhancement of the antioxidant system, as reported for drought-stressed soybean plants by [57]. This suggested that maintaining redox potential could be a necessity for Mexican mint plant protection against drought-induced oxidative stress [58]. Overall, the RH plants had a similar PPP metabolite profile to the RW plants, and the abundance of G6P, 6PG, R5P, Xu5P, G3P, and Se7P in RW, RH, and DHFL plants were similar (Figure 4). Consequently, the concentration of the resultant product, i.e., glucose, in RW, RH, and DHFL plants entering the glycolytic pathway was similar (Figure 3), since PPP runs parallel to glycolysis in the cytosol. Interestingly, the concentration of NADPH in FL and DHFL plants was similarly low as that found in RW plants (Appendix A). The increase in R5P, E4P, and F6P concentrations in FL plants as a result of the flooding, suggests stress tolerance mechanisms can lead to enhanced production of downstream compounds, including nucleotides, aromatic amino acids, and fatty acids [16,59,60]. Overall, Mexican mint plants increased PPP metabolic flux under flooding stress to achieve energy homeostasis—a mechanism reported for *K. candel* by Pan et al. [37].

#### 3.3.4. Tricarboxylic Acid Cycle Intermediates

It was found that both total and individual TCA cycle intermediate metabolites were significantly (*p* < 0.002) affected by the different watering regimes (Table 1; Figure 5). Total TCA cycle metabolites were significantly (*p* < 0.001) increased by ca. 58% in DR plants and ca. 36% in DHFL plants compared to their RW counterparts (Table 1). In contrast, drought stress reduced TCA cycle metabolites in maize, rice, and sesame (*Sesamum indicum*) plants [32,61,62]. This contrasting report could be attributed to differences in plant genotype, i.e., Mexican mint versus maize, rice, and sesame. The fewest total TCA cycle metabolites were found in RH plants and were not significantly (*p* > 0.05) different from those of the RW and FL plants. When compared to RW plants, flooding stress did not alter the total concentration of the TCA metabolites in the Mexican mint plants. This conformed with previous reports for flood-tolerant soybean [63,64], *Acanthus ilicifolius* [65], *Lotus japonicus* [66], and rice [67] plants. According to Menezes-Silva et al. [68], plants can intensify their defense against stress by maintaining information from previous stress events. This “stress memory” in plants involves physiochemical processes such as photosynthesis, maintenance of water status, and osmotic adjustment [69]. These can explain the results of the present study, which showed that the removal of flooding (i.e., DHFL plants) increased total TCA cycle metabolites compared to FL or RW plants (Table 1). It therefore seemed obvious that the ability of DHFL plants to withstand future stress was enhanced compared to FL and RW plants. Although the reductions in TCA cycle metabolites did not significantly (*p* > 0.05) impact ATP production in this study, such reductions could likely alter TCA-mediated amino acid production [70], which must be investigated in future studies. 

More specifically, there was a remarkable increase in eight out of nine of the individual TCA cycle intermediate metabolites in the FL plants (Appendix A) compared to the other treatments. The concentrations of incoming PA from glycolysis and the TCA cycle intermediate metabolites—isocitric acid, α-ketoglutaric acid, succinic acid, fumaric acid, and malic acid—were particularly high in FL plants compared to all the other treatments (Figure 5). This suggests greater mitochondrial activity in FL plants to generate carbon skeletons for amino acid biosynthesis [70]. The highest concentrations of isocitric acid, aconitic acid, and citric acid were noticed in DR plants, which had the least concentrations of oxaloacetic acid, acetyl-CoA, malic acid, and α-ketoglutaric acid. 

Apart from oxaloacetic acid, which was comparatively high in RW plants, all the other metabolites were low. The low concentrations of succinic acid, isocitric acid, acotinic acid, and citric acid in RW plants were similar to those in RH and DHFL plants (Figure 5). Particularly, the RH plants had low concentrations of all the individual TCA cycle intermediate metabolites. For the DHFL plants, the increase in TCA cycle metabolites can be ascribed to high levels of acetyl-CoA, oxaloacetic acid, and malic acid. According to Sweetlove et al. (2010) [70], TCA cycle intermediate metabolites do not have similar flux, and the activities of the different enzymes in the different steps are independent of each other. As such, the suppression of one enzyme within the cycle does not alter the activity of the other enzymes. Therefore, the high concentrations of TCA cycle metabolites in FL and DHFL plants can be attributed to the link between the TCA cycle and other associated pathways, including ammonium assimilation and the biosynthesis of amino acids, nucleotides, and secondary metabolites that contribute to plant stress tolerance [69,70].

Studies showed that changes in malate, citrate, α-ketoglutaric acid, and fumarate during flood stress occurred in many different plants [63,64,66]. The accumulation of succinate is obvious during flood-induced hypoxia conditions since succinate dehydrogenase requires oxygen [52]. Additionally, both drought and flood stresses stimulate the accumulation of citrate in plants [23,64]. In addition, citrate is not only involved in amino acid metabolism but also serves as an antioxidant and intermediate in respiratory metabolisms to generate energy for defense pathways in stress adaptation mechanisms [71]. Furthermore, α-ketoglutaric acid plays an important role in respiration and N assimilation for the biosynthesis of proline, glutamate, glutamine, and arginine. These amino acids function in regulating osmotic potential and act as osmolytes to maintain protein integrity and to mediate water stress tolerance in plants [3,52]. Therefore, like many other plants, the accumulation of these metabolites can be essential for water stress tolerance in Mexican mint plants. 

#### 3.3.5. Nucleotide Biosynthetic Pathway Intermediates

Apart from guanine monophosphate (GMP), all the determined nucleotide pathway intermediate metabolites were significantly (*p* < 0.05) influenced by variation in watering regime (Table 1; Appendix A). Nucleotide metabolism is the most critical cellular component for plant growth and affects several metabolic processes [20]. These nucleotides are essential for information storage and recovery in dividing and expanding tissues. The results revealed that total nucleotide intermediate concentrations were significantly (*p* < 0.001) highest in RH (ca. 229%) and FL (ca. 162%) plants compared to RW plants (Table 1). Total concentrations of nucleotide intermediates in DR, DHFL, and RW plants were not significantly (*p* > 0.05) different and ranged from 267 to 311 nmol/g. Evidence from several studies revealed that nucleotide biosynthesis increased drought tolerance in plants such as Arabidopsis [72], orchids (*Dendrobium* spp.) [73,74], pitaya (*Hylocereus undatus*) [75], drought-tolerant wild wheat (*Triticum boeoticum*) [76], and soybeans [55]. Based on these previous reports, we surmised that nucleotide metabolism would be a critical mechanism for water stress tolerance in Mexican mint. These nucleotides may also allow for the repair and maintenance of water stress-induced cellular damages, which, as a result, might have promoted water stress tolerance in RH and DHFL plants. 

Furthermore, the *de novo* biosynthesis of purine is characterized by the formation of adenosine monophosphate (AMP) and GMP [20]. AMPs are synthesized from activated ribose (5-phosphoribosyl-1-pyrophosphate), key amino acids such as glutamine and glycine, and 10-formyl tetrahydrofolate [20]. GMPs are obtained from deamination of AMP or direct transport of inosine 5′-monophosphate (IMP) in the chloroplast [20]. The results of the present study showed that the relative concentrations of the individual nucleotide intermediates, i.e., guanine diphosphate (GDP), GTP, AMP, and adenosine diphosphate (ADP), known to be involved in purine biosynthesis, were significantly (*p* < 0.001) increased in DR plants, except for GMP (Figure 6). Conversely, these metabolites were significantly (*p* < 0.05) reduced in the FL plants, except for GMP, which was highest in both FL and RW plants.

During the *de novo* biosynthesis of pyrimidine, uridine monophosphate (UMP) is typically formed from phosphoribosyl diphosphate (PRPP), aspartate, and carbamoylphosphate to play a critical role in cytosine monophosphate (CMP) biosynthesis [20,22]. The results of the present study indicated that concentrations of UMP, uridine triphosphate (UTP), uridine diphosphate (UDP), and CMP metabolites were significantly (*p* < 0.001) increased in DR plants compared to RW plants (Figure 6). Ultimately, UMP, UDP, and CMP concentrations were increased in FL plants, while UTP concentrations were slightly reduced. Intriguingly, purine and pyrimidine nucleotides may also function as co-substrates for carbohydrate metabolism. For example, ADP-glucose is an activated precursor for starch synthesis, while UDP and UTP are directly involved in the synthesis, degradation, and transportation (i.e., in the form of UDP-glucose or UTP-glucose) of various forms of carbohydrate, including sucrose, sugar components of glycoproteins, and cell wall matrix polysaccharides [22,77]. UDP-glucose acts as a glucosyl donor for the biosynthesis of hormones and secondary metabolites [77]. Furthermore, ADP-glucose and UDP-glucose/UDP-mannose concentrations in the Mexican mint plants were significantly (*p* < 0.05) elevated in FL plants but were not altered in DR plants compared to the RW plants (Figure 6). These findings suggested that carbohydrate metabolism and secondary metabolite syntheses can enhance flood tolerance. This provides a new insight into drought and flooding tolerance in Mexican mint plants.

### 3.4. Association between the Central Carbon Metabolism Routes 

A multivariate analysis using 2-D PCA biplots and the Pearson correlation coefficient (r) was used to further demonstrate the nature of the association amongst the different routes involved in CCM as influenced by variation in watering regime. The PCA biplot in Figure 7 showed a projection of the response variables in the factor space, which explained ca. 96% of the total variations in the dataset. It was found that total metabolites in the Calvin cycle had a significantly (*p* < 0.001) strong positive association with those of the TCA cycle (r = 0.81) and PPP (r = 0.75) intermediate metabolites. Total PPP intermediate metabolites had a significantly (*p* < 0.01) moderately positive association with the TCA cycle intermediate metabolites (r = 0.68), and a negative correlation with total glycolysis intermediate metabolites (r = −0.70; *p* < 0.005; Table 2).

Furthermore, total nucleotide biosynthetic metabolites had a significant (*p* < 0.01) moderately positive correlation with total glycolysis metabolites (r = 0.65) and a moderate non-significant (*p* > 0.05) association (r = 0.50) with total Calvin cycle metabolites (Table 2). Moreover, since the Calvin cycle, PPP, and glycolytic pathways have some metabolites in common, the association amongst the various individual pathway metabolites involved in the CCM routes showed a distinct and significant (*p* < 0.05) positive association (Appendix A). 

The results in Figure 8 confirmed that most of the specific Calvin cycle intermediate metabolites had a significant (*p* < 0.05) positive association with major TCA cycle intermediate metabolites. For instance, Se7P, R5P, and Ru5P concentrations positively correlated with fumaric acid and succinic acid concentrations, whereas RuBP and xylose-5-phosphate (X5P) exhibited a moderate and strong positive association with aconitic acid and citric acid, respectively. Additionally, PPP intermediate metabolites exhibited a significant (*p* < 0.05) positive correlation with most of the TCA cycle metabolites (Figure 8; Appendix A). In addition to Se7P, R5P, and Ru5P, 6-phosphogluconate exhibited a strong positive association with aconitic acid and citric acid. Glucose had a strong positive association with isocitric acid, aconitic acid, and citric acid, while PA—a key product of glycolysis—had a strong positive association with fumaric acid, succinic acid, α-ketoglutaric acid, isocitric acid, and citric acid (Figure 8; Appendix A). These associations with respect to Mexican mint plants agree with the report by Rocha et al. [66]. They explained that the products of glucose metabolism feed the TCA cycle for energy generation and the synthesis of other metabolic compounds for other pathways. Additionally, ATP and NADPH, which are essential for energy production in plants during water stress adaptation, had a moderately negative correlation with malic acid, fumaric acid, and succinic acid and a moderately positive association with aconitic acid and citric acid (Figure 8; Appendix A). 

The present study showed moderately positive associations between glucose and AMP, ADP, GDP, GTP, CMP, and UTP, and strong positive associations between glucose and UDP and UDP-glucose (Figure 8). Additionally, PA had moderate associations with ribose, AMP, GMP, CMP, ADP-glucose, and UDP-glucose. Similarly, key individual Calvin cycle intermediate metabolites, i.e., RuBP, Ru5P, and R5P, showed strong positive associations with the different nucleotide biosynthetic metabolites.

Overall, the coordinated association between the various pathway metabolites confirmed that water stress instigated metabolic reprogramming of Mexican mint plants to regulate energy production and the biosynthesis of essential compounds necessary for water stress tolerance and adaptation. Additionally, the PCA biplot showed that flooding had a strong influence on the Calvin cycle and TCA cycle, which suggested the possibility of modulating nucleotide biosynthesis and the PPP pathway. On the contrary, drought stress tends to strongly affect glycolysis and moderately influence nucleotide metabolism, with less influence on the Calvin cycle and TCA cycle. This is consistent with previous studies that revealed stimulation of glucose metabolism in plants during drought as a key strategy for energy production to maintain basic cellular functions and thereby mediate drought tolerance [3,23,46]. Additionally, flood-tolerant plants undergo anaerobic respiration and drive an increase in photosynthetic carbon production via the Calvin cycle [37]. In addition, the function of the TCA cycle is to support ATP synthesis for stress tolerance. Furthermore, dehydration after flooding showed a strong influence on the PPP pathway and a moderate effect on the Calvin cycle and the TCA cycle. Nevertheless, rehydration after drought had only a moderate influence on the PPP pathway and no influence on the other CCM routes. 

## 4. Conclusions

Global climate change and a shift in weather patterns have made water stress one of the major limiting factors for crop growth and productivity worldwide. In this study, we demonstrated that water stress affects CCM routes associated with plant resilience and growth. The determined 68 key metabolites involved in the CCM routes, i.e., the Calvin cycle, glycolysis, tricarboxylic acid cycle, pentose phosphate pathway, and nucleotide biosynthesis, were remarkably influenced by both flooding and drought and their respective reversals. This indicates that the accumulation of intermediate metabolites associated with these CCM routes is crucial for energy production and provides carbon skeletons for further macromolecule biosynthesis during internal metabolic reprogramming for water stress tolerance and adaptation. However, plants undergo a fast recovery process following rehydration of drought plants or dehydration of flooded plants to normal growth conditions with no obvious effect on the CCM routes, although it is impacted. For the first time, we revealed the metabolic alteration of Mexican mint plants under different watering regimes and elucidated the mechanism of its water stress tolerance, which can be extended to other members of the Lamiaceae family. We recommend further analysis using transcriptomic and proteomic approaches to identify genes and proteins that regulate these CCM routes in response to water stress. 

## Figures and Tables

**Figure 1 metabolites-13-00539-f001:**
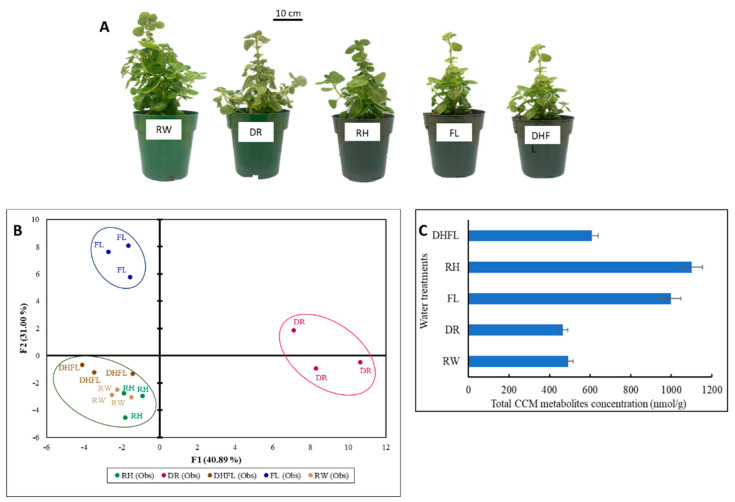
Response of Mexican mint (*Plectranthus amboinicus*) plants to varying watering regimes. (**A**) Plant growth, (**B**) two-dimensional principal component analysis of metabolic profiles, and (**C**) a bar chart of total concentrations of intermediate metabolites along the central carbon metabolic (CCM) routes with error bars. Regular watering (RW), drought (DR), flooding (FL), resumption of regular watering after flooding (DHFL) or after drought (RH).

**Figure 2 metabolites-13-00539-f002:**
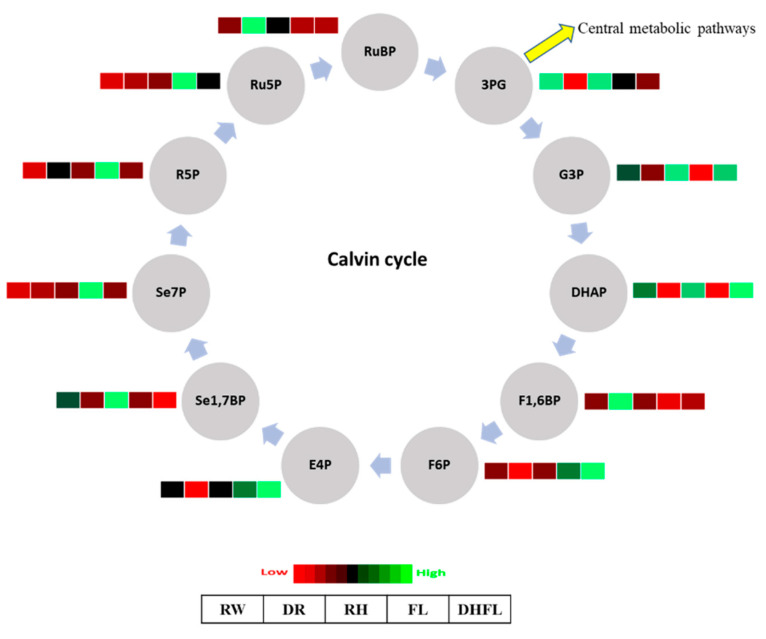
A heat map depicting the individual concentrations of intermediate metabolites of the Calvin cycle in Mexican mint (*Plectranthus amboinicus*) plants under varying watering regimes (n = 3). 3-phosphoglyceric acid (3PG), glyceraldehyde-3-phosphate (G3P), dihydroxyacetone phosphate (DHAP), fructose-1,6-bisphosphate (F1,6BP), fructose-6-phosphate (F6P), erythrose-4-phosphate (E4P), sedoheptulose-1,7-bisphosphate (Se1,7BP), sedoheptulose-7-phosphate (Se7P), ribose-5-phosphate (R5P), ribulose-5-phosphate (Ru5P), and ribulose-1,5-bisphosphate (RuBP). Metabolite concentrations in each block compartment are normalized across all data for an individual compound such that similar color intensities between compounds can represent widely differing concentrations. The red color represents a lower concentration, and the green color represents a higher concentration of a particular metabolite. The compartments were arranged from left to right as follows: regular watering (RW), drought (DR), resumption of regular watering after drought (RH), flooding (FL), and resumption of regular watering after flooding (DHFL).

**Figure 3 metabolites-13-00539-f003:**
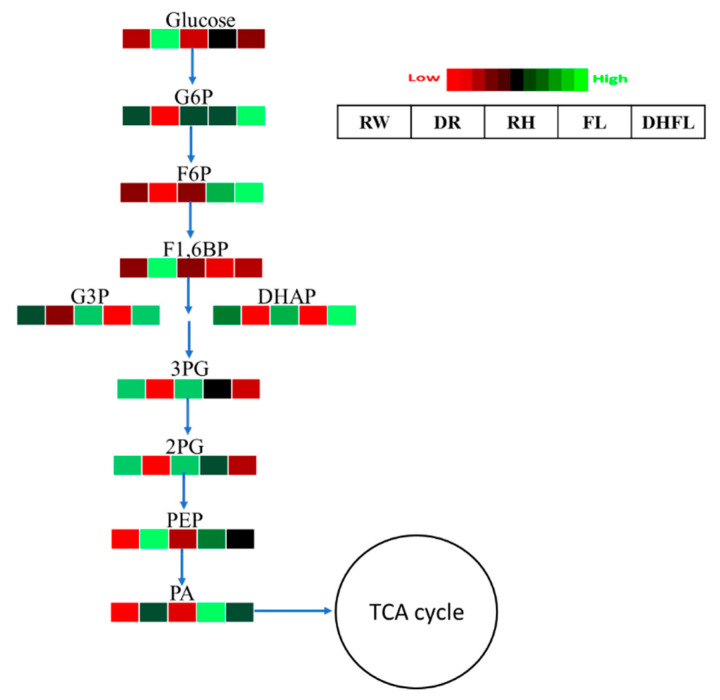
A heat map depicting the individual concentrations of intermediate metabolites of the glycolysis pathway in Mexican mint (*Plectranthus amboinicus*) plants under varying watering regimes (n = 3). Glucose-6-phosphate (G6P), fructose-6-phosphate (F6P), fructose-1,6-bisphosphate (F1,6BP), glyceraldehyde-3-phosphate (G3P), dihydroxyacetone phosphate (DHAP), 3-phosphoglyceric acid (3PG), 2-phosphoglyceric acid (2PG), phosphoenolpyruvic acid (PEP), and pyruvic acid (PA). Metabolite concentrations in each block compartment are normalized across all data for an individual compound such that similar color intensities between compounds can represent widely differing concentrations. The red color represents a lower concentration, and the green color represents a higher concentration of a particular metabolite. The compartment was arranged from left to right as follows: regular watering (RW), drought (DR), resumption of regular watering after drought (RH), flooding (FL), and resumption of regular watering after flooding (DHFL).

**Figure 4 metabolites-13-00539-f004:**
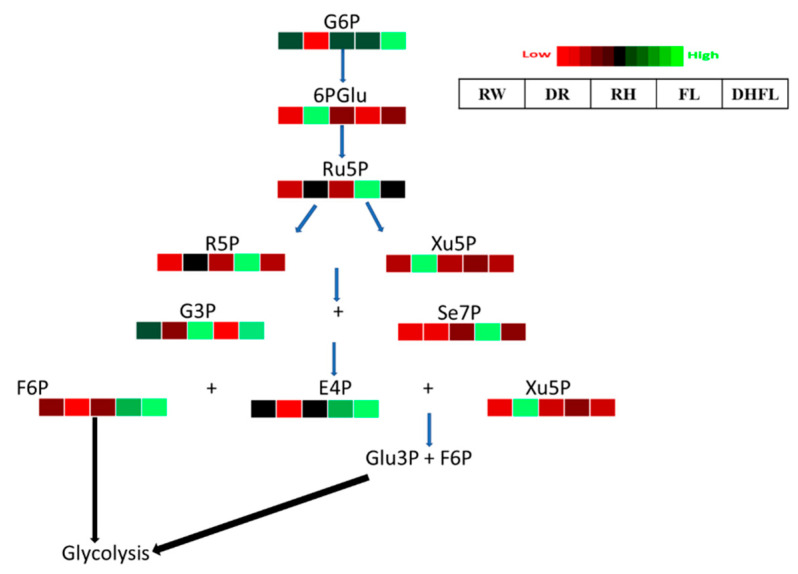
A heat map depicting the individual concentrations of intermediate metabolites of the pentose phosphate pathway in Mexican mint (*Plectranthus amboinicus*) plants under varying watering regimes (n = 3). Glucose-6-phosphate (G6P), 6-phosphogluconate (6PGlu), ribulose-5-phosphate (Ru5P), ribose-5-phosphate (R5P), xylulose-5-phosphate (Xu5P), glyceraldehyde-3-phosphate (G3P), sedoheptulose-7-phosphate (Se7P), fructose-6-phosphate (F6P), erythrose-4-phosphate (E4P), and glucose-3-phosphate (Glu3P). Metabolite concentrations in each block compartment are normalized across all data for an individual compound such that similar color intensities between compounds can represent widely differing concentrations. The red color represents a lower concentration, and the green color represents a higher concentration of a particular metabolite. The compartment was arranged from left to right as follows: regular watering (RW), drought (DR), resumption of regular watering after drought (RH), flooding (FL), and resumption of regular watering after flooding (DHFL).

**Figure 5 metabolites-13-00539-f005:**
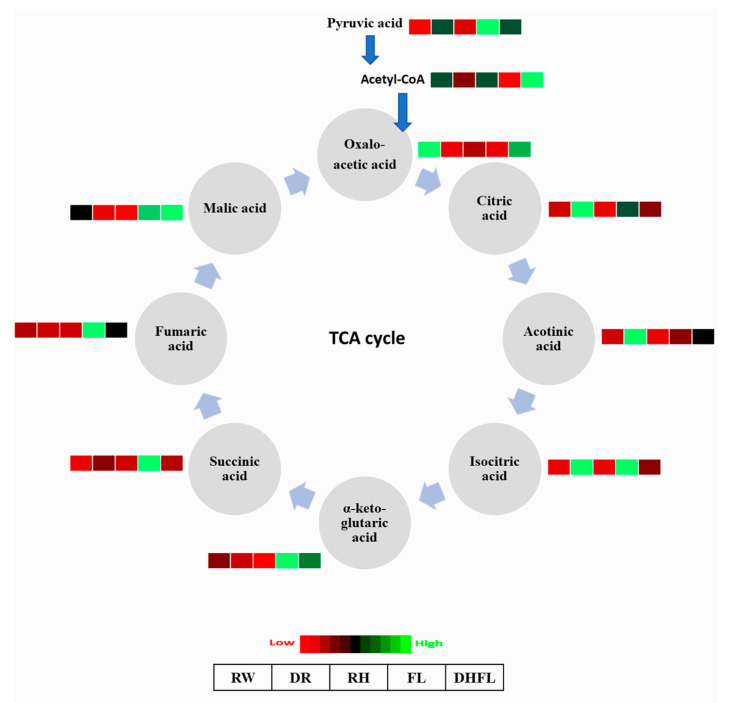
A heat map depicting the individual concentrations of intermediate metabolites of the tricarboxylic acid cycle in Mexican mint (*Plectranthus amboinicus*) plants under varying watering regimes (n = 3). Metabolite concentrations in each block compartment are normalized across all data for an individual compound such that similar color intensities between compounds can represent widely differing concentrations. The red color represents a lower concentration, and the green color represents a higher concentration of a particular metabolite. The compartments were arranged from left to right as follows: regular watering (RW), drought (DR), resumption of regular watering after drought (RH), flooding (FL), and resumption of regular watering after flooding (DHFL).

**Figure 6 metabolites-13-00539-f006:**
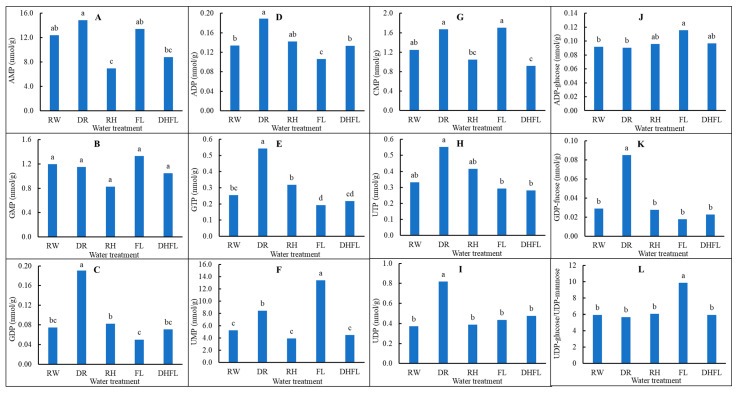
Changes in nucleotide biosynthetic pathway metabolites in Mexican mint (*Plectranthus amboinicus*) plants in response to varying watering regimes (n = 3). (**A**) Adenosine monophosphate (AMP), (**B**) guanine monophosphate (GMP), (**C**) guanine diphosphate (GDP), (**D**) adenosine diphosphate (ADP), (**E**) guanine triphosphate (GTP), (**F**) uridine monophosphate (UMP), (**G**) cytosine monophosphate (CMP), (**H**) uridine triphosphate (UTP), (**I**) uridine diphosphate (UDP), (**J**) ADP-glucose, (**K**) GDP-fucose, and (**L**) UDP-glucose/UDP-mannose. Different alphabetical letters on the bars denote significant differences according to Tukey’s honestly significant difference post-test analyses at a significant level of *p* < 0.05.

**Figure 7 metabolites-13-00539-f007:**
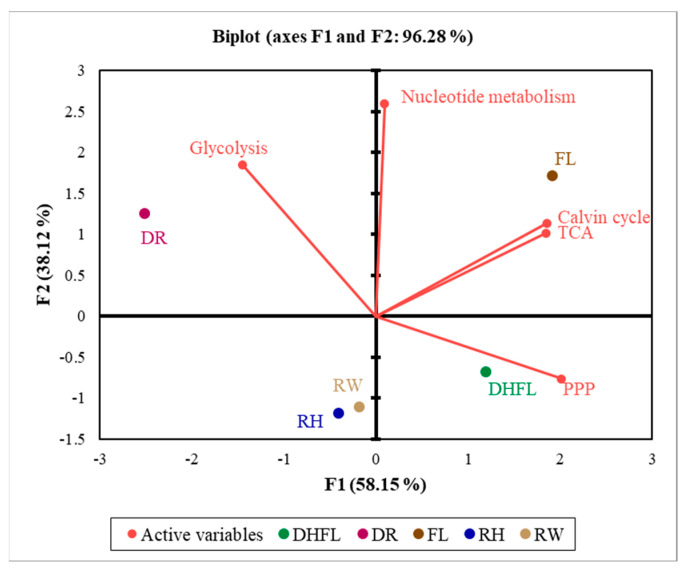
A two-dimensional principal component analysis biplot showing relationships amongst the total metabolites involved in the Calvin cycle, glycolysis, pentose phosphate pathway (PPP), tricarboxylic acid (TCA) cycle, and nucleotide biosynthetic pathway of Mexican mint (*Plectranthus amboinicus*) plants under varying watering regimes. The projection of the variables in the 2D factor space (F1 and F2) explained a total of ca. 96% of the variations in the dataset. Variables that are closely located are not different compared to variables located at a distance within a quadrant or between quadrants (n = 3). Regular watering (RW), drought (DR), resumption of regular watering after drought (RH), flooding (FL), and resumption of regular watering after flooding (DHFL).

**Figure 8 metabolites-13-00539-f008:**
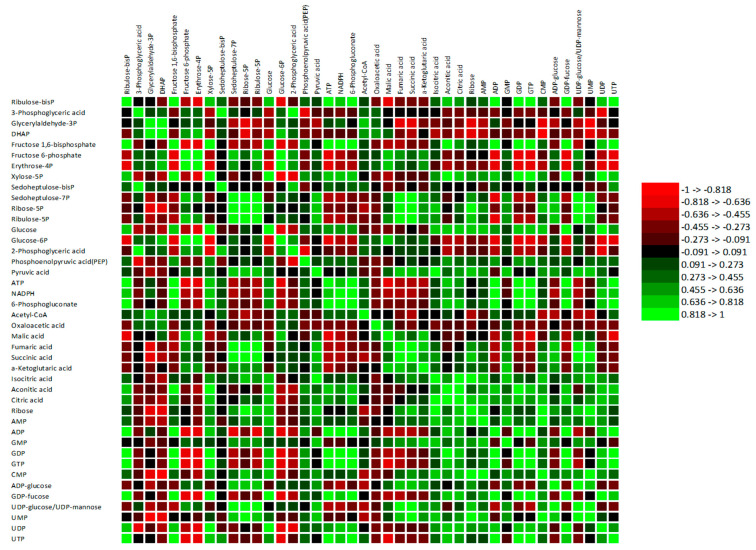
A heat map of the composite correlation matrix between individual metabolites of the central carbon metabolic routes in Mexican mint (*Plectranthus amboinicus*) plants in response to varying watering regimes (n = 3). The red color represents a strong negative association, and the green color represents a strong positive association. Adenosine triphosphate (ATP), adenosine monophosphate (AMP), guanine monophosphate (GMP), guanine diphosphate (GDP), adenosine diphosphate (ADP), guanine triphosphate (GTP), uridine monophosphate (UMP), cytosine monophosphate (CMP), uridine triphosphate (UTP), uridine diphosphate (UDP), ADP-glucose, GDP-fucose, and UDP-glucose/UDP-mannose.

**Table 1 metabolites-13-00539-t001:** Total metabolites involved in specific central carbon metabolic routes in Mexican mint (*Plectranthus amboinicus*) plants under varying watering regimes.

Treatment	Calvin Cycle (nmol/g)	Glycolysis (mmol/g)	TCA (mmol/g)	PPP (nmol/g)	Nucleotide Biosynthesis (nmol/g)
DR	31.33b	29.42a	59.86a	56.67b	288.65b
FL	67.75a	9.04b	36.62bc	182.62a	701.27a
DHFL	47.61ab	2.57c	51.62ab	196.32a	310.62b
RH	41.06b	2.07c	27.59c	149.38a	879.76a
RW	36.95b	1.90c	37.83bc	147.73a	267.60b
*p*-value	0.002	0.00	0.002	0.000	0.000

PPP, pentose phosphate pathway; TCA, tricarboxylic acid. Regular watering (RW), drought (DR), flooding (FL), resumption of regular watering after flooding (DHFL) or after drought (RH). Values are means of three replications, and different alphabetical letters denote significant differences according to Tukey’s honestly significant difference post-test analyses at a significant level of *p* < 0.05.

**Table 2 metabolites-13-00539-t002:** Pearson correlation coefficients (r) amongst the specific central carbon metabolic pathways in Mexican mint (*Plectranthus amboinicus*) under varying watering regimes at a significance level of *p* ≤ 0.05.

	TCA	Glycolysis	Calvin Cycle	PPP
Glycolysis (nmol/g)	r = −0.268 *p* = 0.335			
Calvin cycle (nmol/g)	r = 0.814 *p* = 0.000	r = −0.215 *p* = 0.442		
PPP (nmol/g)	r = 0.684 *p* = 0.005	r = −0.702 *p* = 0.004	r = 0.749 *p* = 0.001	
Nucleotide metabolism (nmol/g)	r = 0.380 *p* = 0.162	r = 0.645 *p* = 0.009	r = 0.501 *p* = 0.057	r = −0.168 *p* = 0.549

PPP, pentose phosphate pathway; TCA, tricarboxylic acid cycle.

## Data Availability

The data presented in this study are available on request from the corresponding author. The data are not publicly available due to privacy.

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
