# Peer review of "How Central Carbon Metabolites of Mexican Mint (Plectranthus amboinicus) Plants Are Impacted under Different Watering Regimes"

_metabolites, 2023, doi:10.3390/metabo13040539_

Round 1

Reviewer 1 Report

1- How much drought is there i.e., soil moisture content must be presented otherwise the experiment doesnot look scientific

2- Similarly, in flooding stress the soil moisture content must be recorded.

3- How many biological and technical replicates were taken are not been mentioned.

4- Figure 8 doesnot clearly state that which stress is been correlated.

5- Because of several flaws in the study the conclusions are not clear.

4-

Author Response

Dear Reviewer,

We would like to thank you very much for your suggestions/comments.

The detailed responses to your comments/suggestions is attached.

Thanks

Reviewer 2 Report

This is a systematic metabolomic study on the shifts of the five main central metabolism pathways in the leaves of Mexican mint plants as affected by  long term water stress in two opposite directions – drought or flooding, and the subsequent recovery from stress. Drought and recovery are more studied in terms of metabolite adaptation contrary to flooding stress and recovery. An impressive amount of data on central metabolites was analysed, with properly driven conclusions. The Introduction is clear, concise and well-focused. My major remarks are on MMs. The experimental design is not clearly described. How the treatments were applied? Please give more details about the water regimes. How growth parameters and water status were affected, did the authors analyze any stress parameters or - only metabolites? In the Abstract, the only comment on plant physiology was “Leaf cluster formation and leaf greening were swift following resumption of regular watering”. Plant growth could be appreciated only from fig. 1A – picture of individual treated plants. How samples were taken – from a particular leaves or from the whole leaf mass?  It is important as there could be differences across leaf canopy. Were leaf samples biological replicates? Results and discussion are combined. What is missing for me is the comparison on CAM vs C3 plant metabolic responses under water stress and recovery. Minor remark – line 123 – fluorescence light

Author Response

(The authors gave the same response as above.)

Reviewer 3 Report

The manuscript can be accepted in its present form for publication in Metabolites.

Author Response

(The authors gave the same response as above.)

Reviewer 4 Report

The manuscript entitled "How central carbon metabolites of Mexican mint (Plectranthus amboinicus) plants are impacted under drought and flooding  conditions, and their reversals to regular watering" is well established and gives an insight about Mexican mint plants under flooding and drought conditions.

Author Response

(The authors gave the same response as above.)
